# Reducibility of higher-order networks from dynamics

Maxime Lucas [1,2,3] ✉, Luca Gallo [4,5,6], Arsham Ghavasieh [7], Federico Battiston [4,8,11] & Manlio De Domenico [7,9,10,11]

Empirical complex systems can be characterized not only by pairwise interactions, but also by higher-order (group) interactions influencing collective phenomena, from metabolic reactions to epidemics. Nevertheless, higher-order networks' apparent superior descriptive power—compared to classical pairwise networks—comes with a much increased model complexity and computational cost, challenging their application. Consequently, it is of paramount importance to establish a quantitative method to determine when such a modeling framework is advantageous with respect to pairwise models, and to which extent it provides a valuable description of empirical systems. Here, we propose an information-theoretic framework, accounting for how structures affect diffusion behaviors, quantifying the entropic cost and distinguishability of higher-order interactions to assess their reducibility to lower-order structures while preserving relevant functional information. Empirical analyses indicate that some systems retain essential higher-order structure, whereas in some technological and biological networks it collapses to pairwise interactions. With controlled randomization procedures, we investigate the role of nestedness and degree heterogeneity in this reducibility process. Our findings contribute to ongoing efforts to minimize the dimensionality of models for complex systems.

Many complex systems exhibit an interconnected structure that can be encoded by pairwise interactions between their constituents. Such pairwise interactions have been used to model biological, social and technological systems, providing a powerful descriptive and predictive framework[1–6]. Recently, the analysis of higher-order structural patterns and dynamical behaviors attracted the attention of the research community as a powerful framework to model group interactions[7–10], with applications ranging from neuroscience[11,12] to ecology[13] and social sciences[14,15], highlighting the emergence of novel phenomena and non-trivial collective behavior[16–21].

Higher-order networks encode more information than pairwise interactions: for example, metabolic reactions are more realistically described by group interactions between any number of reagents and reactants, capturing information that would be lost by considering the union of pairwise interactions instead. However, this modeling flexibility comes at a cost: new data needs to be adequately recorded and stored as group interactions instead of pairwise ones, and new

[1]Department of Mathematics and Namur Institute for Complex Systems (naXys), Université de Namur, Namur, Belgium. [2]Mycology Laboratory, Earth and Life Institute, Université Catholique de Louvain, Louvain-la-Neuve, Belgium. [3]CENTAI Institute, Turin, Italy. [4]Department of Network and Data Science, Central European University, Vienna, Austria. [5]ANETI Lab, Corvinus Institute for Advanced Studies (CIAS), Corvinus University, Budapest, Hungary. [6]Center for Social Data Science (SODAS), University of Copenhagen, Copenhagen, Denmark. [7]Department of Physics and Astronomy "Galileo Galilei", University of Padua, Padova, Italy. [8]Department of AI, Data and Decision Sciences, Luiss University of Rome, Viale Romania 32, 00197 Rome, Italy. [9]Padua Center for Network Medicine, University of Padua, Padova, Italy. [10]Istituto Nazionale di Fisica Nucleare, Sez. Padova, Padova, Italy. [11]These authors contributed equally: Federico Battiston, Manlio De Domenico. ✉e-mail: maxime.lucas@unamur.be

analytical[22–25] and computational[26–28] tools need to be developed. Moreover, the complexity and computational cost of these tools increase exponentially as larger group interactions are considered.

It is therefore crucial to understand under which conditions higher-order representations need to be favored over classical pairwise ones, and whether it is possible to devise a quantitative procedure to determine which representation provides a suitable description of an empirical system.

A research line approaches higher-order interactions from the perspective of "behavior", seeking to characterize statistical dependencies in multivariate time series[29–33]. Within this framework, information-theoretic methods have shown that such dependencies can be decomposed into contributions from different interaction orders[34]. At the same time, initial studies have revealed that the relationship between higher-order behaviors and *mechanisms*, i.e., structure and dynamics, is subtle and often non-trivial[10,35]. How to reduce a higher-order structure (and the dynamics it supports) given the dynamics on top of it, remains an open problem.

A similar challenge was faced nearly a decade ago, when the advent of temporal and categorical data[36–38] allowed multilayer representations of complex networks[39]. For these representations, an information-theoretic approach was used to show that not all layers, or types of interaction, are equally informative: Some information can be discarded or aggregated to reduce the overall complexity of the model without sacrificing its descriptive power[40,41]. Although multilayer networks are different from higher-order networks, this approach, formally based on the density matrix formalism[42], provides a good candidate for the present case. Indeed, the idea is similar in spirit to the widely used information compression algorithms adopted in computer science: by exploiting the regularities in the data, one can build a compressed representation that optimizes the number of bits—i.e., the description length—needed to describe the data with a model and those to encode the model itself. Broadly speaking, this minimum description length principle, grounded on Bayesian inference, allows one to maximize model accuracy while minimizing model complexity by means of a principled regularization term[43–45]. Similar approaches have also been used to coarse-grain complex and multiplex networks[46–48].

Here, we build on this long-standing research line and propose an approach to compress systems with higher-order interactions, while accounting for the ability to describe complex data and the cost due to the complexity of the model. Specifically, we determine a suitable order up to which interactions need to be considered to obtain a reduced and functionally plausible representation of the system. Larger orders above this threshold can be safely discarded. Formally, we do so by generalizing the concept of network density matrix[42,49] to account for higher-order diffusive dynamics with the multiorder Laplacian[17,50] and deriving a meaningful rescaling of the diffusion time as a function of the order of interaction. Then, we calculate an information-theoretical cost function which depends on the largest order considered. By minimizing this cost function, we find the optimal compression of the data which, in turn, corresponds to the most parsimonious functional description of the system, at a given diffusion time. In the following, we refer to this procedure as functional reduction. A higher-order network is fully reducible—to a pairwise network—if the optimal order is 1, while its reducibility decreases for increasing optimal orders. To this aim, we consider two alternative, yet complementary, perspectives: the first, based on the Kullback-Leibler divergence $D_{KL}(\boldsymbol{\rho}|\boldsymbol{\rho_0})$, quantifies the entropic cost of building a system represented by a density matrix $\boldsymbol{\rho}$ from a collection of isolated nodes represented by a density matrix $\boldsymbol{\rho_0}$; the second, based on the Jensen-Shannon divergence $D_{JS}(\boldsymbol{\rho}|\boldsymbol{\rho_0})$, measures how distinguishable the system is from such a collection. These two measures are complementary alternatives, each with distinct advantages and limitations in capturing reducibility, as they emphasize different informational aspects of the system. The choice between them depends on the specific application and context. For simplicity, in the following we present results based on the former, and refer to the Supplementary Information for the analysis using the latter.

We analytically derive the cost function for a rotationally invariant structure, revealing how reducibility depends on Laplacian eigenvalues and diffusion time. Then, we demonstrate the validity of our method by performing an extensive analysis of synthetic networks and investigate the functional reducibility of a broad spectrum of real-world higher-order systems. Our analysis reveals (i) that datasets from different domains are differently reducible, (ii) that no simple structural metric alone can explain reducibility, and (iii) the role of degree heterogeneity in reducibility by using randomization procedures.

The advantage of this framework is that it provides a bridge between network analysis and information theory by means of a formalism that is largely inspired by quantum statistical physics, which has found a variety of applications from systems biology[51] to neuroscience[52], shedding light on fundamental mechanisms such as the emergence of network sparsity[53] and the renormalization group[54].

## Results

### Density matrix for higher-order networks

The flow of information between nodes in a (pairwise) network can be modeled by different dynamical processes. However, in the absence of specific knowledge or hypotheses about the underlying dynamics, diffusion represents the maximum-caliber estimate and thus provides a principled modeling choice[53]. Diffusion on networks can be described by means of the propagator $e^{-\tau\mathbf{L}}$, where $\mathbf{L}$ the combinatorial Laplacian and $\tau$ is the diffusion time. In particular, the information flow from node $i$ to node $j$ is described by the component $\left(e^{-\tau\mathbf{L}}\right)_{ij}$. Network states can then be encoded by a density matrix[42,49] defined as

$$\boldsymbol{\rho}_\tau = \frac{e^{-\tau\mathbf{L}}}{Z}, \qquad (1)$$

where the partition function $Z = \mathrm{Tr}(e^{-\tau\mathbf{L}})$ ensures a unit trace for this operator. The influence through the network guided by the propagator is often described as information flow. However, we note that this should not be conflated with information flow in the information-theoretic sense, as quantified by entropy, mutual information, or transfer entropy. Here, the term is rather used in the broader network science convention to indicate the spread of dynamical effects through the system.

Here, we generalize density matrices to higher-order networks. The most general formalism to encode higher-order networks is that of hypergraphs[8]. A hypergraph is defined by a set of nodes and a set of hyperedges that represent the interactions between any number of those nodes. A hyperedge is said to be of order $d$ if it involves $d + 1$ nodes: a 0-hyperedge is a node, a 1-hyperedge is a 2-node interaction, a 2-hyperedge is a 3-node interaction, and so on. Simplicial complexes are a special case of hypergraph that is also commonly used: they additionally require that each subset of each hyperedge is included in the hypergraph, too. In a hypergraph $H$ with maximum order $d_{max}$, diffusion between nodes through hyperedges of order up to $D \le d_{max}$ can be described by the rank−2 projected operator known as multiorder Laplacian[50,55]

$$\mathbf{L}^{[D]} \equiv \mathbf{L}^{(D,\mathrm{mul})} = \sum_{d=1}^{D} \frac{\gamma_d}{\langle K^{(d)} \rangle} \mathbf{L}^{(d)}, \qquad (2)$$

which is a weighted sum of the Laplacians $\mathbf{L}^{(d)}$ at each order $d$ up to order $D$. At each order, the weight is defined by a real coefficient $\gamma_d$ (which we set to 1 for simplicity) and the averaged generalized degrees $\langle K^{(d)} \rangle$. Each $d$-order Laplacian is defined by $L_{ij}^{(d)} = K_i^{(d)}\delta_{ij} - \frac{1}{d}A_{ij}^{(d)}$, in

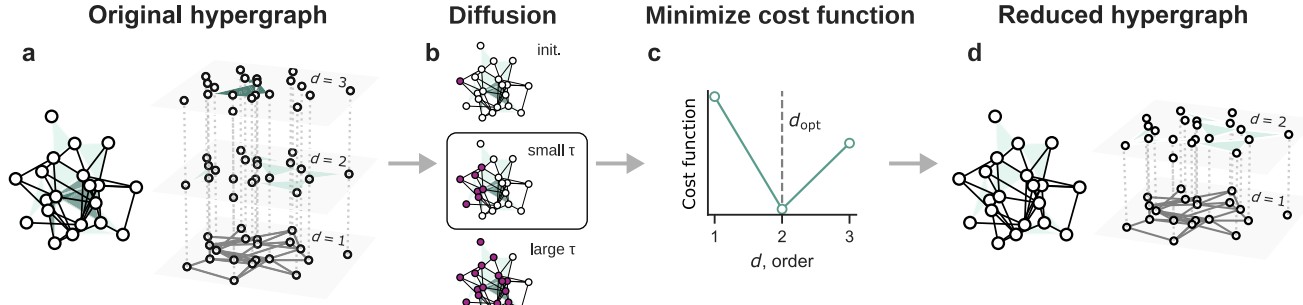

**Fig. 1 | Functional reducibility of higher-order networks.** Illustration of our method with an example hypergraph. Given **a** an original hypergraph with interactions of orders up to $d_{max}$ (=3, here), **b** we use higher-order diffusion processes to probe the system at a chosen diffusion time $\tau$ (e.g., small) that acts as a topological scale. **c** We then compute the cost function, a trade-off between information loss and model complexity, of the same hypergraph, but considering orders only up to $d$. We determine the optimal order $d_{opt}$ as the one that minimizes the cost function. Finally, **d** we reduce the original hypergraph to an optimal version by considering orders up to $d_{opt}$.

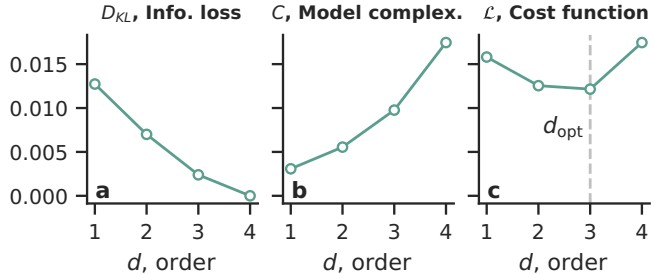

**Fig. 2 | The cost function is the sum of information loss and model complexity based on Kullback–Leibler divergence.** We illustrate the terms in the cost function defined in Eq. (4), on an example random simplicial complex: **a** information loss $D_{KL}(\rho_\tau^{[d_{max}]}|\rho_{\tau'}^{[d]})$, **b** model complexity $C(\rho_\tau^{[d]})$, and **c** their sum, the cost function $\mathcal{L}(\rho_\tau^{[d_{max}]}|\rho_{\tau'}^{[d]})$. The minimum of the cost function is indicated by the vertical line. Parameters were set to $N = 100$ nodes and wiring probabilities $p_d = 50/N^d$ at order $d$ with $d_{max} = 4$. See S.I. Sec. V for results based on Jensen–Shannon divergence.

terms of the generalized degrees $K^{(d)}$ and adjacency matrix $\mathbf{A}^{(d)}$ of order $d$. The matrix $\mathbf{L}^{[D]}$ satisfies all the properties expected from a Laplacian: it is positive semidefinite and its rows (columns) sum to zero (see Methods Section I for details). In general, a hypergraph does not need to have hyperedges at every order below $D$, unless it is a simplicial complex. If there is no hyperedge of order $d$, both the Laplacian and the average degree in Eq. (2) vanish, and the result is undefined. In those cases, the sum thus needs to be taken over all orders below $D$ that exist: $\mathcal{D} = \{d \leq D : \langle K^{(d)} \rangle > 0\}$.

The multiorder density matrix of hypergraph $H$, up to order $d$, is defined as

$$\rho_\tau^{[d]} = \frac{e^{-\tau\mathbf{L}^{[d]}}}{Z}, \qquad (3)$$

with the partition function $Z = \mathrm{Tr}(e^{-\tau\mathbf{L}^{[d]}})$. Just like its pairwise analog in Eq. (1), this operator satisfies all the expected properties of a density matrix: it is positive definite, and its eigenvalues sum up to one. Importantly, the diffusion time $\tau$ plays the role of a topological scale parameter: small values of $\tau$ allow information to diffuse only to neighboring nodes, probing only short-scale structures. Larger values of $\tau$, instead, allow the diffusion to reach more remote parts of the hypergraph and describe large-scale structures. In this context, the meaning of "small" and "large" depends on the network structure, and can be estimated with respect to the magnitude of the largest ($1/\lambda_{max}$) and smallest (non-zero) ($1/\lambda_{min}$) eigenvalues of the Laplacian matrices, respectively. To meaningfully compare the density matrix in Eq. (3) at

different orders $d$, we will need to rescale the chosen $\tau$ at each order—the derivation and necessity of the rescaling are detailed below in Section C.

## Quantifying the reducibility of a hypergraph

We approach the reducibility of a hypergraph as a problem of model selection. We formulate that problem as follows: given a hypergraph $H$ with maximum order $d_{max}$, is $H$ an optimal representation of itself, or is considering only its hyperedges up to a given order $d < d_{max}$ sufficient? Formally, we treat the density matrix $\rho^{[d_{max}]}$ as data and $\rho^{[d]}$ as a model of the data. Formulated in this way, we need to find the "optimal" model of the data, that is, to evaluate the optimal order $d_{opt}$. To this aim, it is desirable to frame the optimization problem in Bayesian terms, following the minimum description length principle, akin to Occam's razor. That framework quantifies the suitability of a model through its likelihood and its complexity through the prior. However, one important limitation should be noted in our case: defining a prior in terms of the density matrix is non-trivial, preventing the definition of a unique and self-consistent complexity term as in the original minimum description length principle. To address this, we introduce suitable—though not necessarily optimal—cost functions. Another important aspect is that the likelihood function provides only a proxy description of the observed network, expressed through the diffusive process unfolding on it. This requires interpreting our results in terms of the interplay between structure and dynamics, rather than structure alone.

The steps of the methods are illustrated in Fig. 1: (a) start from an original hypergraph, (b) use higher-order diffusion at a selected diffusion time $\tau$, then (c) calculate the optimal largest order $d_{opt}$ by minimizing the cost function, and (c) reduce the original hypergraph to a suitable hypergraph, that is, one with orders only up to $d_{opt}$ without losing functional information.

As illustrated in Fig. 2, we define the cost function $\mathcal{L}$ as the sum of the information loss—the opposite of the model accuracy—of the model, measured by a suitably generalized Kullback–Leibler divergence $D_{KL}$, and the model complexity $C$ (see "Methods" Section K for details):

$$\mathcal{L}\left(\rho_\tau^{[d_{max}]}|\rho_\tau^{[d]}\right) = D_{KL}\left(\rho_\tau^{[d_{max}]}|\rho_{\tau'}^{[d]}\right) + C\left(\rho_{\tau'}^{[d]}\right), \qquad (4)$$

where $\tau'$ is a necessary order-dependent rescaling of $\tau$, as will be explained in Section C.

Note that by definition, there is no information loss when considering all possible orders, i.e., $D_{KL}\left(\rho_\tau^{[d_{max}]}|\rho_\tau^{[d_{max}]}\right) = 0$. It is worth remarking that this cost function deviates from the standard Bayesian description length for probability distribution, because it is not trivial to define the corresponding prior term in terms of operators (see

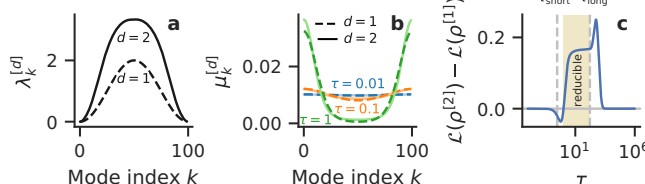

**Fig. 3 | Analytical reducibility of hyperrings. a** Eigenvalues of the multiorder Laplacians up to order $d = 1, 2$, for a hyperring with $N = 100$ nodes. **b** Eigenvalues of the corresponding density matrices, for $\tau = 0.01$ (blue), 0.1 (orange), and 1 (green), for $d = 1$ (dashed) and $d = 2$ (solid). **c** Difference between the cost function at order $d = 2$ and $d = 1$ as a function of $\tau$: it is negative (positive) when $d_{opt} = 2$ (=1), i.e., the hypergraph is irreducible (reducible, beige shade). Vertical dashed lines indicate the short ($\tau_{short} = 1/\lambda_{max}^{[2]}$) and long ($\tau_{long} = 1/\lambda_{min}^{[2]}$) timescales of the full system.

"Methods"). To partially overcome this limitation, we use again a Kullback-Leibler divergence to approximate the model complexity as $C(\rho_\tau^{[d]}) = D_{KL}(\rho_\tau^{[d]}|\rho_{iso}) = \log N - S_\tau^{[d]}$, where $S_\tau^{[d]} = -\text{Tr}(\rho_\tau^{[d]} \log \rho_\tau^{[d]})$ is the von Neumann entropy of the hypergraph up to order $d$, and $\rho_{iso}$ the density matrix of $N$ isolated nodes, which has maximal entropy $S_{iso} = \log N$ (see "Methods" Section L). The complexity can take values in $C(\rho_\tau^{[d]}) \in [0, S_{iso}]$, reaching its minimum for hypergraphs with maximal entropy (e.g., isolated nodes or hypergraphs with high regularity), and its maximum for hypergraphs with minimal entropy (e.g., hypergraphs with more complex structures).

Another definition, grounded on quantifying distinguishability, is discussed in Section M and described in S.I. Sec. V.

In the following, we will refer to optimal order $d_{opt}$, at diffusion time $\tau$, as the most suitable order that minimizes our cost function:

$$d_{opt}^{(\tau)} = \text{argmin}_d \mathcal{L}\left(\rho_\tau^{[d_{max}]}|\rho_{\tau'}^{[d]}\right). \tag{5}$$

It is also worth remarking that the optimal order should be interpreted in terms of the diffusion dynamics on the network, not directly its structure. With this prescription, the likelihood term—encoded by $D_{KL}$—in the cost function retains all the features of a likelihood function[42] and can be suitably used for our purpose.

Finally, we define the reducibility of the hypergraph at diffusion time $\tau$ as

$$\chi_\tau(H) = \frac{d_{max} - d_{opt}^{(\tau)}}{d_{max} - 1}, \tag{6}$$

which measures the ratio between the number of orders to reduce and the maximum number of orders to reduce, $d_{max} - 1$. By construction, $\chi_\tau(H) = 0$ for a hypergraph that is not reducible at all, i.e., $d_{opt}^{(\tau)} = d_{max}$, while $\chi_\tau(H) = 1$ for a hypergraph that is maximally reducible, that is, it can be optimally reduced to its pairwise interactions, $d_{opt}^{(\tau)} = 1$. The dependence on $\tau$ makes this a multiscale method: each diffusion time serves as a different lens to probe the system. There is typically a range of informative $\tau$ values determined by the eigenvalues of the multiorder Laplacian, as we will see in Section D.

## Rescaling the diffusion time $\tau$ at each order

As mentioned above, a given diffusion time $\tau$ may be large for some networks but low for others, which is a challenge when the aim is to compare networks and orders. To overcome this problem, we rescale $\tau$ to ensure an appropriate diffusion time for each structure, and we exploit examples of hypergraphs with certain regularities to define a baseline and characterize the scaling relation.

Specifically, there is a class of hypergraphs for which Laplacians of different orders are proportional, i.e., $\mathbf{L}^{(d)} \propto \mathbf{L}^{(d')}$. This occurs in very regular structures, including complete hypergraphs and some simplicial complex lattices (see "Methods" Section N). In this case, since the

Laplacian matrices govern the flow of information, all orders and all their combinations encode the same functional information. Consequently, we expect these structures to be functionally invariant under reduction—i.e., the cost function should not change as one reduces the hypergraph.

However, without rescaling, the summation of Laplacian matrices according to Eq. (2) simply strengthens the flow pathways in the original hypergraph, making it different from its reduced versions. To correct for this effect, we rescale $\tau$ to allow meaningful comparisons between a hypergraph at different orders.

Since the density matrix only depends on the product of $\tau\mathbf{L}^{(d)}$, this can be achieved by selecting a value of diffusion time $\tau$, and then rescaling it to obtain a new $\tau'(d)$ at each order like so:

$$\tau'(d) = \frac{d_{max}}{d}\tau, \tag{7}$$

which we denote $\tau' \equiv \tau'(d)$ to avoid cluttering the notation. This rescaling ensures that the multiorder density matrices are all equal,

$$\rho_{\tau'(d)}^{[d]} = \rho_\tau^{[d_{max}]} \forall d, \tag{8}$$

in the special case of hypergraphs with proportional Laplacian matrices, and gives a flat cost function $\mathcal{L}\left(\rho_\tau^{[d_{max}]}|\rho_{\tau'}^{[d]}\right) = C\left(\rho_\tau^{[d_{max}]}\right)$ for all $d$ in those extreme structures (S.I. Sec. I).

For illustration purposes, we set $\tau = \tau_{short} = 1/\lambda_{max}$ in numerical experiments, unless otherwise stated, where $\lambda_{max}$ is the largest eigenvalue of the multiorder Laplacian of the original hypergraph $\mathbf{L}^{[d_{max}]}$. This corresponds to choosing a small diffusion time to explore more local structures of the hypergraph.

Physically, this rescaling simply means that we adapt the diffusion time at which we probe the hypergraph as we consider more orders, to highlight the distribution of flow pathways rather than their accumulated strengths. We use this rescaling of $\tau$ at each order in all hypergraphs.

## Analytically tractable case: the hyperring

For arbitrary hypergraphs, the cross-entropy part of the information loss is hard to derive analytically. Here, we consider a case that is analytically tractable because of its rotational symmetry: a hyperring[56] with interactions up to order 2.

We define the hyperring as $N$ nodes on a ring with pairwise interactions with first neighbors, $(i, i+1) \mod N, \forall i$ and triplet interactions with first neighbors too $(i-1, i, i+1) \mod N, \forall i$. By construction, this hyperring is a simplicial complex. Because of the rotational symmetry, the Laplacian and density matrices are circulant matrices, which commute, and for which the eigenvalue spectrum is known and expressed in the Fourier basis[56,57]. Specifically, the multiorder Laplacian spectra are (see full derivation in Supp. Info. Sec. II), up to order 2:

$$\lambda_k^{[1]} = 1 - \cos\left(\frac{2\pi k}{N}\right), \tag{9}$$

$$\lambda_k^{[2]} = 2 - \frac{5}{3}\cos\left(\frac{2\pi k}{N}\right) - \frac{1}{3}\cos\left(\frac{4\pi k}{N}\right), \tag{10}$$

with $\lambda_0^{[d]} = 0$ and $\lambda_k^{[d]} = \lambda_{N-k}^{[d]}$ for $0 < k < \lfloor (N-1)/2 \rfloor$ and $d = 1, 2$ (Fig. 3a). We also know that the spectra of the corresponding density matrices can be expressed in terms of these as $\mu_k^{[d]} = e^{-\tau'\lambda_k^{[d]}}/Z^{[d]}$, which depend on $\tau$ (Fig. 3b). These eigenvalues sum to 1 and can be seen as probabilities, or weights associated with each diffusion mode. For short diffusion times, these weights are distributed uniformly (blue curves), but at longer diffusion times, the fast modes decay, and only the slow modes remain (green curves).

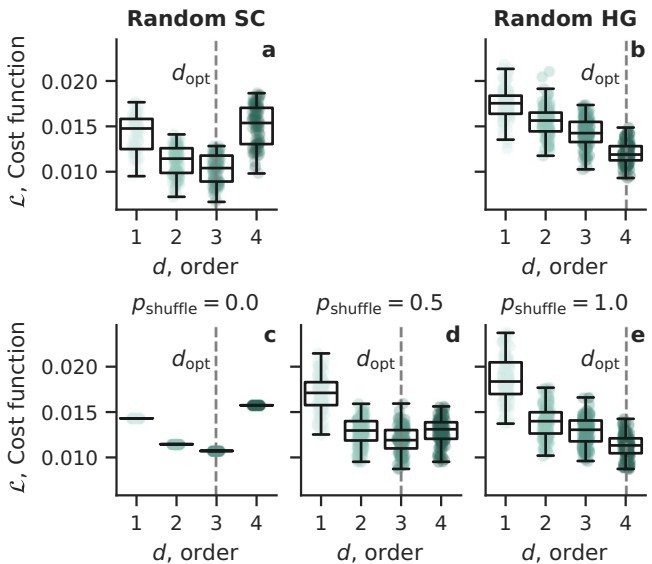

**Fig. 4 | Reducibility of random higher-order networks.** Cost function in **a** random simplicial complexes and **b** random hypergraphs. Cost function for **c** a single random simplicial complex, and then after randomly shuffling **d** 50%, and **e** all of its hyperedges, corresponding to a random hypergraph. From (**c**) to (**e**), the nestedness between hyperedges of different orders decreases. Each point corresponds to one of 100 realizations. Box plots show median (center line), interquartile range (box), and the non-outlier range (whiskers). The $d_{opt}$ minimizing the cost function is indicated by the vertical line. Parameters were set to $N = 100$ nodes and wiring probabilities $p_d = 50/N^d$ at order $d$ with $d_{max} = 4$.

We can now explicitly write the information loss: indeed, it reduces to a KL divergence over the distributions of eigenvalues $\mu_k^{[d]}$ because the matrices are circulant and hence commute. Eventually, we can write

$$D_{KL}\left(\boldsymbol{\rho}_\tau^{[d_{max}]}|\boldsymbol{\rho}_{\tau'}^{[d]}\right) = \tau\left[\left(\frac{d_{max}}{d} - 1\right)\mathbb{E}_{\rho^{[d_{max}]}}\left[\lambda_k^{[d]}\right]\right.$$
$$\left. - \mathbb{E}_{\rho^{[d_{max}]}}\left[\lambda_k^{\lfloor d\rfloor}\right]\right]$$
$$+ \log\left(\frac{Z^{[d]}}{Z^{[d_{max}]}}\right), \qquad (11)$$

where we denote

$$\mathbb{E}_{\rho^{[d_{max}]}}\left[\lambda_k\right] = \sum_{k=0}^{N-1}\mu_k^{[d_{max}]}\cdot\lambda_k, \qquad (12)$$

the expectation of a given spectrum weighted by the diffusion modes under the full system diffusion, and

$$\lambda_k^{\lfloor d\rfloor} = \sum_{\delta=d+1}^{d_{max}}\frac{1}{\left\langle K^{(\delta)}\right\rangle}\lambda_k^{(\delta)}, \qquad (13)$$

the eigenvalues of the hypergraph composed by all truncated orders. Eq. (11) is interpretable and has an overall structure similar to those derived for pairwise networks[42], The first two terms are proportional to $\tau$ and account for "spectral distortion": (1) contributed by orders up to $d$, and (2) not contributed by orders $d < \delta \leq d_{max}$ not present in the truncated system. The $(\frac{d_{max}}{d} - 1)$ pre-factor accounts for the rescaling of $\tau$ and vanishes when $d = d_{max}$. The third term is a standard log-correction of the spread of the two density matrix spectra. This information loss vanishes in the limits $\tau \to 0$ and $\tau \to \infty$ (in the former, the expected values

vanish because diffusion peaks in the uniform mode $k = 0$, the only one not decaying). This indicates that these extreme diffusion times are not informative scales at which to probe the system: at $\tau = 0$ no diffusion has occurred, whereas at $\tau = \infty$ all diffusions have reached the uniform equilibrium. At intermediate $\tau$, the information loss vanishes for $d = d_{max}$, as expected.

We can now derive the complexity in terms of those same eigenvalues:

$$C(\boldsymbol{\rho}_{\tau'}^{[d]}) = \log\left(\frac{N}{Z^{[d]}}\right) - \tau\frac{d_{max}}{d}\,\mathbb{E}_{\rho^{[d]}}\left[\lambda_k^{[d]}\right]. \qquad (14)$$

Note that this expression of the complexity is valid for any hypergraph, as the symmetry of the hyperring was not needed to derive it. The complexity tends to zero, $C(\boldsymbol{\rho}_{\tau'}^{[d]}) \to 0$ for $\tau \to 0$, and $C(\boldsymbol{\rho}_{\tau'}^{[d]}) \to \log N$ for $\tau \to \infty$.

The cost function is simply the sum of Eqs. (11) and (14). In order to obtain the optimal order as a function of the diffusion time $\tau$, we can simply compute the difference between the cost function at orders 2 and 1 (Fig. 3c): the hyperring is irreducible at short diffusion times ($d_{opt} = 1$) but reducible at intermediate and long diffusion times ($d_{opt} = 2$). The sign of the result indicates whether the optimal order is 1 or 2. Note again that extremely short $\tau \to 0$ or long $\tau \to \infty$ diffusion times are not informative scales at which to probe the system.

These analytical expressions are interpretable and reveal how reducibility is directly linked to the structure and diffusion dynamics on top of it via the diffusion time, and the eigenvalues of the multiorder Laplacian. Explicitly, the terms $\mathbb{E}_{\rho^{[d_{max}]}}\left[\lambda_k^{[d]}\right]$, $\mathbb{E}_{\rho^{[d_{max}]}}\left[\lambda_k^{\lfloor d\rfloor}\right]$ quantify the spectral distortion between the truncated and the full hypergraph. At $\tau \sim \tau_{short}$, all modes contribute nearly equally, so reducibility depends on whether higher-order edges introduce genuinely new local features into the spectrum. At $\tau \sim \tau_{long}$, only the slow modes dominate, making reducibility hinge on whether higher orders reshape the global connectivity patterns. In both regimes, the trade-off with model complexity determines whether the gain in descriptive power outweighs the penalty of keeping additional orders. Hence, whenever adding higher-order edges does not substantially alter the relevant spectral content at the probed scale, the system will be deemed reducible. This interpretation also explains why the hyperring appears reducible at most (not-too-small) diffusion times: the added higher-order edges do not substantially alter global patterns of diffusion, because of the rotational symmetry and nestedness—strong structural constraints. From this point of view, the hyperring provides a conceptual bridge between the complete hypergraph, and the more complex, and less symmetric, structures that we investigate numerically in the next sections.

## Random structures

We now investigate the reducibility of two types of heterogeneous random structures: random hypergraphs and random simplicial complexes. To do so, we compute the optimal order—remarking that it refers to the most suitable one according to our cost function—numerically as described above.

A random hypergraph is defined by a number of nodes $N$ and a set of wiring probability $p_d$ for each order required. At each order $d$, a hyperedge is created for any combination of $d + 1$ nodes with probability $p_d$, similarly to Erdős-Rényi networks. Random simplicial complexes are built in the same manner before adding the missing subfaces of all simplices, to respect the condition of inclusion. In both cases, we set $N = 100$, $p_d = 50/N^d$ and $d_{max} = 4$.

Figure 4 shows the cost function considering orders from 1 to 4, for 100 realizations of each type of random structure, at $\tau = \tau_{short}$. In the case of random simplicial complexes (Fig. 4a), $d_{opt} = 3$, reflecting some reducibility $\chi = 1/3$. Instead, in random hypergraphs (Fig. 4b), the

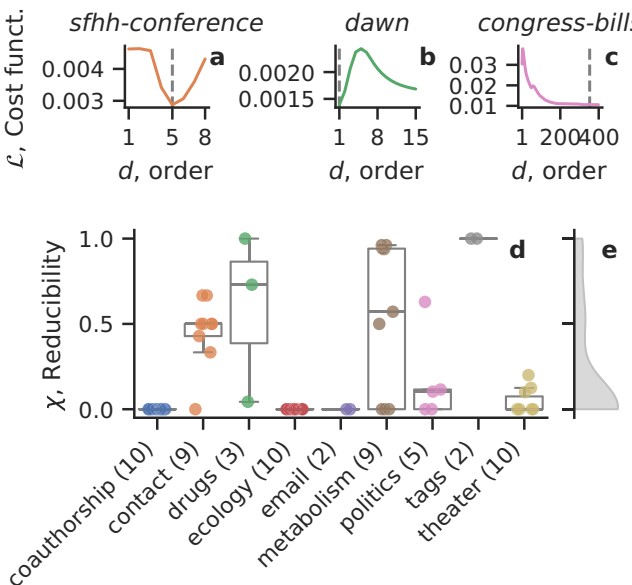

**Fig. 5 | Sixty empirical datasets show different levels of reducibility.** We show the cost function against the largest order considered in **a** `sfhh-conference`, **b** `dawn`, and **c** congress bills. **d** Reducibility $\chi$ for all 60 datasets colored by category and **e** its overall distribution. The number of datasets in each category is reported in parenthesis. Box plots show median (center line), interquartile range (box), and the non-outlier range (whiskers). All reducibility values are reported in Table 1.

optimal order is the maximum, $d_{\mathrm{opt}} = 4 = d_{\max}$. This means that those random hypergraphs, at this scale, are not reducible, i.e., $\chi = 0$.

The crucial difference between these two cases is that the hyperedges between different orders are correlated (nested) in random simplicial complexes but uncorrelated in random hypergraphs. Indeed, simplicial complexes need to satisfy downward inclusion: if a $d$-hyperedge is present, all (lower order) hyperedges that are subsets of its nodes need to be included too. More information about the differences between the two and their impact on dynamics can be found in ref. 17. Nestedness can be measured by the *simplicial fraction* defined in ref. 58: it is 1 (maximal) in simplicial complexes but closer to 0 (minimal) in random hypergraphs.

To test the effect of nestedness on reducibility, we start from a random simplicial complex (Fig. 4c) and gradually change it into a random hypergraph by shuffling its hyperedges. Specifically, we change each hyperedge into another non-existing hyperedge with probability $p_{\mathrm{shuffle}}$, preserving the number of hyperedges. For $p_{\mathrm{shuffle}} = 1$, the result is a random hypergraph (Fig. 4e). Figure 4c–e shows how decreasing nestedness also decreases reducibility.

In simplicial complexes, higher-order hyperedges are strongly nested within lower-order ones. Therefore, higher-order edges that overlap with lower-order ones add little new spectral content, yielding reducibility. By contrast, in random hypergraphs or when nestedness is destroyed through shuffling, additional hyperedges substantially reshape the spectrum by introducing genuinely new local connections, thereby lowering reducibility.

As mentioned above, $\tau$ acts as a topological scale. Diffusion processes with different values of $\tau$ are thus expected to "see" different structures, which may result in different optimal order and reduced hypergraph. To illustrate this, we compute the cost function curves for the random simplicial complex case, for five values of $\tau$ evenly spaced on a logarithmic scale, and where the second and fourth are $1/\lambda_{\max}$ and $1/\lambda_{\min}$, respectively. Supplementary Fig. S2 shows that as the diffusion time increases (larger $\tau$), the cost function (i) increases overall and (ii) the reducibility decreases to $\chi = 0$. Supplementary Fig. S4 shows the same results for three values of the hyperedge density, where we can

see that for higher densities, the cost function curve becomes much flatter with no clear minimum, suggesting a functional similarity between the large-scale structure at all orders.

We similarly tested the effect of the sole density on the reducibility (Supplementary Figs. S3 and S4). In general, the overall density coefficient does not appear to significantly affect the reducibility.

At long diffusion times, reducibility is generally low because the expectations weight only the slow modes, which capture global connectivity patterns. In this regime, higher-order edges have a stronger effect: unless they are tightly nested within lower-order ones, they rewire long-range connections and shift the smallest eigenvalues, making the truncated spectra diverge from the full one. As a result, simplicial complexes retain some reducibility thanks to their nested structure, but random hypergraphs or shuffled ensembles remain irreducible at long times.

### Real-world hypergraphs display diverse reducibility
We now investigate how reducible real-world hypergraphs are by considering 60 empirical hypergraph datasets from 9 categories: coauthorship, face-to-face contacts, drugs, ecology, email, metabolism, politics, online forum tags, and theater.

First, we observe a great variety in the reducibility values (Fig. 5) and the associated cost function curves (Supplementary Figs. S5 and S6) for $\tau = \tau_{\mathrm{short}}$. Figure 5a–c shows three examples with curves of very different shapes and with different reducibility. All reducibility values are shown in Fig. 5d by category, and their overall distribution covers the full possible range with many values close to 0 (Fig. 5e). Second, we note that datasets from different categories seem to be distinctively reducible: none of the coauthorship, email, theater, and ecology datasets are very reducible ($\chi \approx 0$), whereas the contact, drugs, and metabolism datasets show a range of reducibility values, and the tags datasets appear fully reducible ($\chi = 1$). The variability between the drugs datasets can be understood because the nodes and hyperedges represent different concepts in each case (see "Methods" Section O). A Kruskal-Wallis H-test indicated that the category of the dataset has a statistically significant effect on the reducibility, $p < 10^{-4}$— after excluding the categories with too few ($\leq 5$) datasets. A description of all datasets can be found in "Methods" Section O, and their summary statistics and reducibility can be found in Table 1.

For completeness, we also show the reducibility values for a long $\tau = \tau_{\mathrm{long}}$ in Supplementary Fig. S7. We observe that in most cases, the reducibility drops to zero in that case, as in the synthetic cases. This can be understood as follows: At long diffusion times, the dynamics is governed by the slow Laplacian modes (small eigenvalues), which capture global connectivity patterns. Adding higher-order edges almost always alters these global patterns—by creating shortcuts, bridging distant parts, or biasing flows across large subsets of nodes—so the truncated spectra diverge strongly from the full one. As a result, information loss dominates and the system becomes irreducible.

### Reducibility is not explained by any one simple metric
To better understand the relationship between reducibility and the structure of the hypergraphs, we computed the Pearson correlation coefficient between the reducibility and several typical structural metrics.

In Fig. 6, we show the results for six metrics: density ($M/N$, where $M$ is the number of hyperedges and $N$ the number of nodes), the logarithm of the maximum generalized degree, the logarithm of the spectral radius (that is, the maximum of the absolute eigenvalues of the generalized adjacency matrix), nestedness (as measured by the simplicial fraction[58]), the cross-order degree correlation (between orders 1 and 2), and the heterogeneity of the generalized degree (measured as $(K_{\max} - \langle K \rangle)/K_{\max}$). Note that although all metrics use quantities generalized for higher-order networks, such as degree and

**Table 1 | Reducibility of 60 real-world higher-order networks**

| Dataset | Category | Node | Hyperedge | $N$ | $M$ | $d_{max}$ | $d_{opt}$ | $\chi$ |
|---|---|---|---|---|---|---|---|---|
| coauth-mag-geology_1980 | coauthorship | Author | Co-authors of a paper | 1674 | 245 | 17 | 17 | 0.00 |
| coauth-mag-geology_1981 | coauthorship | Author | Co-authors of a paper | 1075 | 1427 | 28 | 28 | 0.00 |
| coauth-mag-geology_1982 | coauthorship | Author | Co-authors of a paper | 1878 | 222 | 25 | 25 | 0.00 |
| coauth-mag-geology_1983 | coauthorship | Author | Co-authors of a paper | 535 | 233 | 14 | 14 | 0.00 |
| coauth-mag-geology_1984 | coauthorship | Author | Co-authors of a paper | 2773 | 271 | 16 | 16 | 0.00 |
| coauth-mag-history_2010 | coauthorship | Author | Co-authors of a paper | 701 | 322 | 34 | 34 | 0.00 |
| coauth-mag-history_2011 | coauthorship | Author | Co-authors of a paper | 654 | 316 | 43 | 43 | 0.00 |
| coauth-mag-history_2012 | coauthorship | Author | Co-authors of a paper | 673 | 335 | 35 | 35 | 0.00 |
| coauth-mag-history_2013 | coauthorship | Author | Co-authors of a paper | 826 | 104039 | 37 | 37 | 0.00 |
| coauth-mag-history_2014 | coauthorship | Author | Co-authors of a paper | 833 | 54933 | 30 | 30 | 0.00 |
| contact-high-school | contact | Person | People in face-to-face contact | 327 | 1459 | 4 | 3 | 0.33 |
| contact-primary-school | contact | Person | People in face-to-face contact | 242 | 24520 | 4 | 2 | 0.67 |
| hospital-lyon | contact | Person | People in face-to-face contact | 75 | 1824 | 4 | 2 | 0.67 |
| hypertext-conference | contact | Person | People in face-to-face contact | 113 | 2434 | 5 | 3 | 0.50 |
| invs13 | contact | Person | People in face-to-face contact | 92 | 787 | 3 | 2 | 0.50 |
| invs15 | contact | Person | People in face-to-face contact | 217 | 4909 | 3 | 2 | 0.50 |
| malawi-village | contact | Person | People in face-to-face contact | 84 | 431 | 3 | 2 | 0.50 |
| science-gallery | contact | Person | People in face-to-face contact | 410 | 796 | 4 | 4 | 0.00 |
| sfhh-conference | contact | Person | People in face-to-face contact | 403 | 6093 | 8 | 5 | 0.43 |
| dawn | drugs | Drug | Drugs used by a patient | 2290 | 903 | 15 | 1 | 1.00 |
| ndc-classes | drugs | Drug label | Labels associated to a drug | 628 | 547 | 38 | 11 | 0.73 |
| ndc-substances | drugs | Substance | Substances in a drug | 3414 | 6471 | 186 | 186 | 0.00 |
| plant-pollinator-mpl-014 | ecology | Plant species | Plants visited by a pollinator | 28 | 3350 | 9 | 9 | 0.00 |
| plant-pollinator-mpl-015 | ecology | Plant species | Plants visited by a pollinator | 130 | 10541 | 103 | 103 | 0.00 |
| plant-pollinator-mpl-016 | ecology | Plant species | Plants visited by a pollinator | 26 | 145053 | 16 | 16 | 0.00 |
| plant-pollinator-mpl-021 | ecology | Plant species | Plants visited by a pollinator | 90 | 169259 | 24 | 24 | 0.00 |
| plant-pollinator-mpl-034 | ecology | Plant species | Plants visited by a pollinator | 25 | 62 | 20 | 20 | 0.00 |
| plant-pollinator-mpl-044 | ecology | Plant species | Plants visited by a pollinator | 107 | 68 | 24 | 24 | 0.00 |
| plant-pollinator-mpl-046 | ecology | Plant species | Plants visited by a pollinator | 16 | 128 | 15 | 15 | 0.00 |
| plant-pollinator-mpl-049 | ecology | Plant species | Plants visited by a pollinator | 37 | 54 | 23 | 23 | 0.00 |
| plant-pollinator-mpl-057 | ecology | Plant species | Plants visited by a pollinator | 114 | 125 | 36 | 36 | 0.00 |
| plant-pollinator-mpl-062 | ecology | Plant species | Plants visited by a pollinator | 456 | 87 | 156 | 156 | 0.00 |
| email-enron | email | Email address | Email | 143 | 85 | 36 | 36 | 0.00 |
| email-eu | email | Email address | Email | 986 | 71 | 39 | 39 | 0.00 |
| STM_v1_0 | metabolism | Metabolite | Metabolites in a reaction | 1716 | 84 | 66 | 5 | 0.94 |
| e_coli_core | metabolism | Metabolite | Metabolites in a reaction | 72 | 35 | 22 | 22 | 0.00 |
| iAB_RBC_283 | metabolism | Metabolite | Metabolites in a reaction | 342 | 38 | 8 | 4 | 0.57 |
| iAT_PLT_636 | metabolism | Metabolite | Metabolites in a reaction | 738 | 401 | 9 | 5 | 0.50 |
| iEC1356_Bl21DE3 | metabolism | Metabolite | Metabolites in a reaction | 1898 | 73 | 105 | 5 | 0.96 |
| iEK1008 | metabolism | Metabolite | Metabolites in a reaction | 965 | 159 | 105 | 5 | 0.96 |
| iJR904 | metabolism | Metabolite | Metabolites in a reaction | 761 | 52 | 52 | 4 | 0.94 |
| iND750 | metabolism | Metabolite | Metabolites in a reaction | 1059 | 179 | 45 | 45 | 0.00 |
| iYO844 | metabolism | Metabolite | Metabolites in a reaction | 977 | 35 | 62 | 62 | 0.00 |
| congress-bills | politics | Congress member | Members on a bill | 1718 | 21721 | 399 | 353 | 0.12 |
| house-bills | politics | House member | Members on a bill | 1494 | 7818 | 398 | 357 | 0.10 |
| house-committees | politics | House member | Members on a committee | 1290 | 12704 | 80 | 80 | 0.00 |
| senate-bills | politics | Senate member | Members on a bill | 294 | 138742 | 98 | 37 | 0.63 |
| senate-committees | politics | House member | Members on a committee | 282 | 301 | 30 | 30 | 0.00 |

**Table 1 (continued) | Reducibility of 60 real-world higher-order networks**

| Dataset | Category | Node | Hyperedge | N | M | $d_{max}$ | $d_{opt}$ | χ |
|---|---|---|---|---|---|---|---|---|
| tags-math-sx | tags | Tag | Tags associated to a question | 1627 | 254 | 4 | 1 | 1.00 |
| tags-ask-ubuntu | tags | Tag | Tags associated to a question | 3021 | 91 | 4 | 1 | 1.00 |
| hyperbard-a-midsummer-nights-dream | theater | Character | Characters co-present on stage | 28 | 866 | 13 | 13 | 0.00 |
| hyperbard-alls-well-that-ends-well | theater | Character | Characters co-present on stage | 31 | 73 | 11 | 9 | 0.20 |
| hyperbard-antony-and-cleopatra | theater | Character | Characters co-present on stage | 87 | 1143 | 12 | 12 | 0.00 |
| hyperbard-as-you-like-it | theater | Character | Characters co-present on stage | 30 | 1098 | 12 | 12 | 0.00 |
| hyperbard-coriolanus | theater | Character | Characters co-present on stage | 75 | 914 | 11 | 10 | 0.10 |
| hyperbard-hamlet | theater | Character | Characters co-present on stage | 41 | 878 | 11 | 11 | 0.00 |
| hyperbard-macbeth | theater | Character | Characters co-present on stage | 49 | 366 | 9 | 8 | 0.13 |
| hyperbard-othello | theater | Character | Characters co-present on stage | 33 | 2318 | 12 | 12 | 0.00 |
| hyperbard-romeo-and-juliet | theater | Character | Characters co-present on stage | 49 | 1011 | 14 | 14 | 0.00 |
| hyperbard-the-tempest | theater | Character | Characters co-present on stage | 22 | 2112 | 14 | 14 | 0.00 |

We report the category, what a node and a hyperedge represent, the number of nodes N, the number of hyperedges M, the maximum order $d_{max}$, the optimal order $d_{opt}$, and the reducibility χ of a range of higher-order networks from empirical datasets, for $\tau_{short}$. The reducibility values are shown by category in Fig. 5.

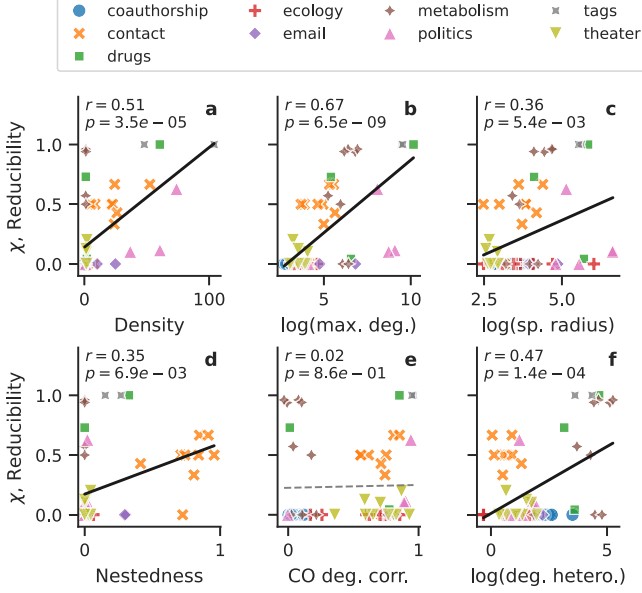

**Fig. 6 | Reducibility is not explained by any one simple metric.** We show the reducibility of the 60 datasets against several structural metrics: **a** density (M/N), **b** log(maximum degree), **c** log(spectral radius), **d** nestedness, **e** cross-order degree correlation, and **f** log(degree heterogeneity). Datasets are colored by category. For each metric, we indicate the Pearson correlation coefficient r, its associated p-value, and show the corresponding linear fit (solid line if significant, dashed line otherwise). Reducibility is significantly linearly correlated to several of those metrics, but for a given value of a metric, reducibility can take many different values.

density, nestedness and cross-order degree correlation are strictly higher-order metrics: they measure how the structure at different orders relate. For each metric, we report the correlation coefficient r and the associated p-value.

Results indicate a significant ($p < 0.05$) positive correlation between reducibility and each of these metrics except for the cross-order degree correlation. However, note that in all cases, for a given value of a structural metric, reducibility still displays a range of values. For example, reducibility is correlated to density ($r = 0.51$, $p = 3.5 \times 10^{-5}$) but density alone cannot explain reducibility: datasets with low density exhibit reducibility values between 0 and 1. Note that this might in part be because most of these metrics only consider the structure aggregated across orders, whereas reducibility considers how the structure and dynamics change as more orders are considered. One exception of this is the nestedness, which accounts for the relation between orders (and cross-order degree correlation, but here only computed between orders 1 and 2 for practical reasons). Hence, if a hypergraph has high nestedness, it gives us information about how the structure changes as larger orders are added.

For completeness, we report the correlation between reducibility and several additional metrics in Supplementary Fig. S8a–f and the correlations between those metrics in Supplementary Fig. S8g.

### Linking reducibility to degree heterogeneity by randomizing

We start by focusing on nestedness as a first metric to help us better understand reducibility in empirical datasets. Nestedness describes how often hyperedges of lower orders are subsets of hyperedges of higher orders. Simplicial complexes are maximally nested because they satisfy closure downwards, whereas random hypergraphs are minimally nested because there is no correlation between its hyperedges at different orders. To quantify nestedness, we use the simplicial fraction as above.

There are at least three reasons to start with nestedness: (i) nestedness is a form of structural redundancy, (ii) in synthetic hypergraphs, we observed that decreasing nestedness in a controlled manner decreased reducibility (Fig. 4), and (iii) in empirical datasets, we observed a positive correlation between nestedness and reducibility.

As noted in ref. 58, many empirical datasets are not very nested. This is also the case in the datasets considered here, with the only nested ones belonging to the "contact" category. For this reason, we

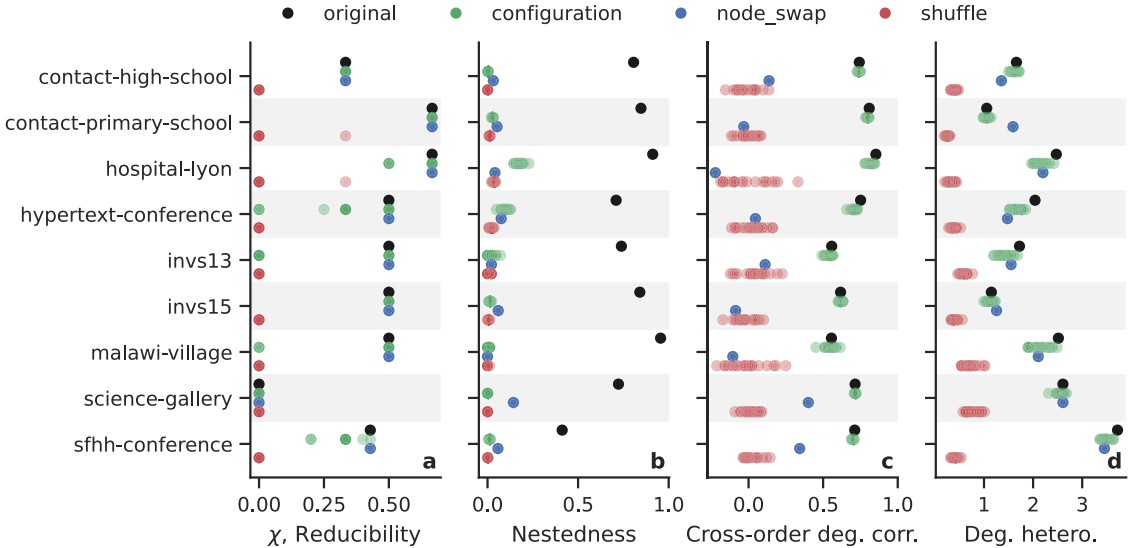

**Fig. 7 | Link between reducibility and degree heterogeneity by randomizing.** We focus on the "contact" datasets because they have high reducibility and nestedness. To each dataset (black), we apply three randomizing strategies: the configuration model (green) a node swap (blue), and a random shuffling of hyperedges (red). In each case, we show **a** the reducibility, **b** the nestedness measured by the simplicial fraction, **c** the cross-order degree correlation between orders 1 and 2, and **d** the degree heterogeneity. Each dot represents one of 20 realizations of each randomizing strategy. Although all randomizing strategies destroy the nestedness, preserving the degree heterogeneity is sufficient to preserve the reducibility.

now focus on that category, which also displays a range of reducibility values (Fig. 5).

To investigate the link between reducibility and nestedness, we use three randomization strategies (from least to most structure-destructive): the configuration model[59], a node swap procedure[17], and a simple shuffling of hyperedges[17] (see "Methods" Section I for details). In the configuration model, we randomize each order of interaction independently. This randomizes hyperedges while preserving the degrees of the nodes at each order. In the node swap procedure, at each order $d > 1$, we pair nodes $(i, j)$ and swap their generalized degrees $K_i^{(d)} \leftrightarrow K_j^{(d)}$ by swapping their hyperedge memberships at that order. This preserves the degree distributions at each order, but destroys the degree sequences (i.e., the assignment of degrees to specific nodes). In the shuffling, we simply shuffle all hyperedges. This randomizes hyperedges too, while only preserving the number of hyperedges—the degree sequences and distributions are destroyed. This shuffling was used in the synthetic random structure (Fig. 4c, d). Compared to the first two randomizing procedures, the shuffling is extreme and yields structures akin to random hypergraphs. Theoretically, we expect these three randomization procedures to destroy nestedness in the datasets since they destroy correlations between orders, and we are interested in seeing their effect on reducibility.

We show the results for 20 realizations of each randomization procedure in Fig. 7. First, we observe that all three procedures destroyed nestedness, as expected (Fig. 7b): original values (black) were mostly 0.7 (and one around 0.4), but nestedness is close to 0 in all structures randomized by the configuration model (green), the node swap (blue), and the shuffling (red). As anticipated, the shuffling also caused the reducibility to vanish (Fig. 7a), consistent with what we observed in synthetic structures (Fig. 4c, d). Interestingly, however, the configuration model and the node swap preserved the reducibility values most of the time. In other words, in hypergraphs with originally large nestedness and reducibility, the reducibility is preserved if the degree sequences and distributions are, too, but is destroyed if not. This indicates that it is the information contained in the degree distributions at each order, rather than the nestedness, that is impacting the reducibility of these systems.

To go further, we examined the cross-order degree correlation and the degree heterogeneity. The first measures how correlated the

degrees of nodes at different orders are (Fig. 7c). It is a summary statistic of the degree sequences (but cannot be computed from the degree distributions, as it contains no information about the degree-node assignments). As a proxy, we simply look at the correlation between orders 1 and 2: $DC = \text{corr}(\{K_i^{(1)}\}, \{K_i^{(2)}\})$. Second, the degree heterogeneity measures how varied the generalized degrees are (Fig. 7d). It is a summary statistic of the degree distributions (and sequences, but the distributions are sufficient). In very nested hypergraphs, "richer nodes get richer" and cross-order degree correlation is high[17]. As expected, the original "contact" datasets have high DC (close to 1) which is destroyed (close to 0) by the shuffling and the node swap. DC is only preserved by the configuration model, that is, by preserving the degree sequences, as expected. This indicates that the DC cannot explain the reducibility in this case, since the reducibility can be preserved when the DC is destroyed (node swap). Then we turn to the generalized degree heterogeneity: it is only destroyed by the shuffling, and thus preserved by preserving degree sequences (configuration model) or distributions (node swap). We thus observe that it is sufficient that the randomization preserves degree heterogeneity to preserve reducibility.

We also checked the effect of randomizations on other metrics: the higher-order degree heterogeneity ratio[17], the spectral radius, and maximum higher-order degree (Supplementary Fig. S9). The maximum degree, another summary statistic of the degree distributions, shows a similar but less marked trend than the degree heterogeneity: it decreases but is not fully destroyed when the shuffling destroys the reducibility. The degree heterogeneity ratio and spectral radius do not exhibit any clear trend in this case.

In conclusion, the results suggest that the degree distributions and their summary statistic of degree heterogeneity (and, to a lesser extent, the maximum degree) are key to the reducibility of these "contact" datasets that are highly nested, whereas nestedness and cross-order degree correlation are not. From the perspective of spectral distortion, this suggests that heterogeneity—which we know is directly linked to the broadness of the Laplacian spectrum—correlates in these datasets with not much new spectral content as orders are added, at short times, thereby enabling lower-order truncations to approximate the full spectrum. Applying other suitable randomizations to other categories of datasets (with lower nestedness) could help disentangle other key ingredients to reducibility. Interestingly,

past research showed that cross-order degree correlation affected dynamics on hypergraphs: low DC stabilizes synchronization[17], and higher DC facilitates bistability contagion models[59].

## Discussion

All areas of natural and social science, as well as engineering ones, are undergoing a deluge of publicly available data with complex structure. This data allows for building more detailed and powerful models of systems from areas such as physics, biology, sociology, and technology. One class of such models is networks that encode group interactions rather than just pairwise ones: higher-order networks. Higher-order networks provide a natural framework for modeling systems where interactions between more than two units occur, such as chemical reactions of metabolic interest or social interactions. However, they are usually projected—by design—to pairwise interactions when data is gathered, and higher-order information is inevitably lost. By preserving that information, higher-order networks have the potential to yield a more reliable model of some empirical systems. Nevertheless, higher-order networks come with novel challenges: new theoretical and computational methods have to be developed, while algorithms become exponentially more complex for increasing order of the interactions. It is thus of paramount importance to understand under which conditions one should opt for higher-order modeling—or keep using traditional pairwise models.

Here, we have provided a method, at the edge between statistical physics and information theory, to guide researchers in identifying the most suitable representation for their data by using diffusion processes. We built this method by adapting the well-tested density matrix formalism to higher-order interactions with the multiorder Laplacian, and the diffusion time serves as a topological scale. A key challenge in doing so was to derive a meaningful scaling of the diffusion time across interaction orders. Our method is based on minimizing a suitable cost function, encoding both the number of bits required to describe the data given a model and the number of bits required to describe the model, estimated by its entropy, to find a suitable order of group interactions. Orders above the optimal one can be safely discarded because they provide only redundant information about the system, while increasing model complexity and the computational cost for its analysis, according to our definitions.

As anticipated, the cost function we define here does not correspond to a true description length. Indeed, density matrices are operators rather than probability distributions, and no direct generalization of Bayes' theorem exists for them. To partially address this limitation, we employ the Kullback-Leibler divergence to approximate model complexity and thereby define an effective regularization term.

Alternative definitions may be appropriate depending on the context and objective. For instance, we also considered a cost function based on the Jensen-Shannon divergence, which quantifies how distinguishable the system—at a given order—is from a gas of isolated nodes relative to the pairwise case. In this case, the results did not consistently support the need for a higher-order description, highlighting that the apparent lack of universality arises from the specific features each cost function is able to capture (Supplementary Information Section V). The choice of definition will thus depend on the specific application, context, and practical use. This points to the need for a more principled approach yet to be established.

Additionally, although we considered here unsigned, unweighted, and undirected hypergraphs, the formalism can be directly generalized to relax all those conditions (resulting in a density matrix that is not a Gibbsian-like exponential), following recent work that has been done so for pairwise network[60].

Remarkably, we show that not all systems require a higher-order model to be described and that even within the same class of systems, there is some level of variability. We started by testing the method on extremely regular cases, where the cost function was flat for all orders,

as computed analytically. We then analytically derived the cost function of a slightly more complex case: the hyperring. This revealed how reducibility is to the structure and dynamics of the system, with an explicit dependence on the diffusion time and the multiorder Laplacian eigenvalues. In particular, spectral distortion, or how diffusion is affected by truncating orders, is key.

We then applied the method to random structures, where results indicate that random hypergraphs are non-reducible: i.e., they are best represented by considering all possible orders of interaction. This is understandable: in these ensembles, hyperedges are maximally uncorrelated both within and across orders, so each order introduces genuinely new spectral content that cannot be compressed without significant information loss. In contrast, random simplicial complexes are reducible because higher-order edges are generated in a correlated fashion from lower-order ones. This correlation induces redundancy in the Laplacian spectrum across orders, so that at a given diffusion scale the dynamics can often be captured without retaining the full multiorder structure.

We then applied our method to empirical datasets that contain group interactions. At short diffusion times, the large variety of reducibility values obtained indicates that while the extra information encoded in higher-order networks provide essential local information that cannot be discarded for some systems, others can be suitably represented with lower orders and in some cases even with only pairwise interactions without substantial loss. The category of system appears to have a significant effect on these short-time reducibility patterns, with contact datasets typically more reducible than, for instance, coauthorship. At long diffusion times, however, most datasets become irreducible, consistent with the fact that slow diffusion modes capture global patterns: here, higher-order edges reshape the large-scale connectivity, and their contribution cannot be neglected. This was observed in general, suggesting that shorter diffusion times are more informative in practice.

A correlation analysis revealed that reducibility is correlated with many simple structural metrics, such as density, maximum degree, nestedness, or degree heterogeneity, but that none of them alone can explain reducibility. To further investigate the link between structural properties and reducibility, we analyzed contact datasets because they have high reducibility. We focused on nestedness, a form of higher-order structural redundancy, cross-order degree correlations, and degree distribution, designing three randomization strategies to subsequently destroy each of these structural features. In this way, we revealed the role of degree distribution rather than nestedness or cross-order degree correlations on reducibility. This is of particular interest, as these three higher-order structural metrics have been linked to changes in dynamics in higher-order networks in the past[17,59]. This is also consistent with the role of the multiorder Laplacian spectra, known to be link to the degrees of the structure.

Our results challenge the widespread assumption that complex network data must necessarily be investigated through the lens of higher-order dynamics. In fact, there are completely reducible and non-reducible systems, with a spectrum of cases between these two extremal cases, demonstrating that some orders might be irrelevant or uninformative to describe an empirical system. Moving forward, it will be crucial to collect new datasets from the biological sciences to determine whether and in what scenarios complex networks such as connectomes can be considered reducible or irreducible. This will help to understand under which conditions higher-order mechanisms and behaviors are essential for the function of these systems, as is often assumed nowadays.

Our work also contributes to the increasing interest in reducing the dimensionality of higher-order network models[61,62], and more generally, of complex systems models[54,63–66]. In particular, phase reduction techniques for coupled oscillators have shown that higher-order interactions appear naturally when reducing the dynamics of

(initially pairwise) coupled higher-dimensional oscillators to simple 1-dimensional phase oscillators[63]. This is consistent with recent results reducing complex to low-rank matrices, in which case higher-order interactions also appear naturally[66]. A renormalization group approach also identified important orders of interaction in empirical datasets which were not always pairwise[62]. Complementary to these and to these approaches that look at "mechanisms"—the structure and dynamical rules controlling the evolution of the system—other approaches such as maximum entropy models and higher-order information-theoretic measures look at "behaviors"—the result of the evolution, that is, time series of observables from those systems. Interestingly, several such studies have instead shown that pairwise models can be sufficient, under certain conditions, to describe the behaviors (or statistics) of systems that contain higher-order interactions, or mechanisms[31–34]. For example, in ref. 32, the authors show that a statistical model truncated to pairwise correlations is sufficient to approximate the full multivariate statistics produced by a system of spins with higher-order interactions, and in ref. 34, the authors decompose the contributions of different orders from discrete random variables. Except for initial work disentangling between higher-order mechanisms and behaviors[10] and showing that the relationship between the two is complex[35], little is known about their relationship, suggesting a promising direction for future work.

Detecting redundancies and exploiting them with an approach to identify a compressed representation of the data has the potential to enhance our understanding of empirical higher-order systems and contributes to the increasing interest in dimensionality reduction of such networks[61,62,66]. The main advantage of our framework is that it builds on a consolidated formalism that is firmly grounded on the statistical physics of strongly correlated systems. As in the case of multilayer systems[40], we think that it is remarkable that it is possible to tackle such challenges by capitalizing on a formal analogy between quantum and higher-order systems, which can be further exploited to gain novel insights into the structural and functional organization of complex systems.

## Methods

### Multiorder Laplacian for hypergraphs

The Laplacian matrix **L** of a graph is defined as $L_{ij} = K_i \delta_{ij} - A_{ij}$, where $K_i$ is the degree of node $i$, $\delta_{ij}$ is the Kronecker delta, and **A** is the adjacency matrix. For hypergraphs, we can define $d_{\max}$ Laplacian matrices, one for each order of interaction. The Laplacian of order $d$ can be defined as[17,50]

$$L_{ij}^{(d)} = K_i^{(d)} \delta_{ij} - \frac{1}{d} A_{ij}^{(d)}, \tag{15}$$

where $K_i^{(d)}$ is the degree of order $d$ of node $i$, i.e., the number of $d$-hyperedges connected to node $i$, while $\mathbf{A}^{(d)}$ is the adjacency matrix of order $d$, whose elements $A_{ij}^{(d)}$ counts the number of $d$-hyperedges connected to nodes $i$ and $j$.

We can hence define the multiorder Laplacian up to an order $D$ as[17,50]

$$\mathbf{L}^{[D]} = \mathbf{L}^{(D, \text{mul})} = \sum_{d=1}^{D} \frac{\gamma_d}{\langle K^{(d)} \rangle} \mathbf{L}^{(d)}, \tag{16}$$

where $\gamma_d$ is a tuning parameter of interactions of order $d$, while $\langle K^{(d)} \rangle$ is the average degree of order $d$. For simplicity, in this study, we always set $\gamma_d = 1$.

Note that in general, a hypergraph does not need to have hyperedges at every order below $D$, unless it is a simplicial complex. If there is no hyperedge of order $d$, both the Laplacian of order $d$ and the average degree in Eq. (2) vanish and the result is undefined. In those cases, the sum thus needs to be taken over all orders below $D$ that exist: $\mathcal{D} = \{d \leq D : \langle K^{(d)} \rangle > 0\}$.

### Multiorder density matrix for hypergraphs

We define the multiorder density matrix of a hypergraph $H$, up to order $d$, as

$$\boldsymbol{\rho}_\tau^{[d]} = \frac{e^{-\tau \mathbf{L}^{[d]}}}{Z}, \tag{17}$$

with the partition function $Z = \text{Tr}(e^{-\tau \mathbf{L}^{[d]}})$ and the diffusion time $\tau$. As its pairwise analog in Eq. (1), this operator satisfies all the expected properties of a density matrix: it is positive definite, and its eigenvalues sum up to one.

We know from ref. 42 how the eigenvalues of the density matrix $\lambda_i(\boldsymbol{\rho}_\tau^{[d]})$ are related to those of the Laplacian $\lambda_i(\mathbf{L}^{[d]})$:

$$\lambda_i(\boldsymbol{\rho}_\tau^{[d]}) = \frac{e^{-\tau \lambda_i(\mathbf{L}^{[d]})}}{Z} \tag{18}$$

where the partition function can also be expressed as $Z = \sum_{i=1}^{N} e^{-\tau \lambda_i(\mathbf{L}^{[d]})}$. This relation is useful to link our intuition and knowledge about the Laplacian spectrum to that of the density matrix.

The diffusion time $\tau$ plays the role of a topological scale parameter: small values of $\tau$ allow information to diffuse only to neighboring nodes, probing only short-scale structures. Larger values of $\tau$, instead, allow the diffusion to reach more remote parts of the hypergraph and describe large-scale structures. In this context, the meaning of small and large depends on the network structure, and can be estimated with respect to the magnitude of the largest ($1/\lambda_{\max}$) and smallest ($1/\lambda_{\min}$) eigenvalues of the Laplacians, respectively.

### Information loss as Kullback–Leibler divergence between two hypergraphs

The state of the original hypergraph $H$ is stored in the multiorder density matrix

$$\boldsymbol{\rho}_\tau^{[d_{\max}]} = e^{-\tau \mathbf{L}^{[d_{\max}]}} / Z^{[d_{\max}]}, \tag{19}$$

and the state of the reduced hypergraph where we consider orders up to $d$ is given by

$$\boldsymbol{\rho}_\tau^{[d]} = e^{-\tau \mathbf{L}^{[d]}} / Z^{[d]}. \tag{20}$$

To perform model selection and determine the optimal order $d_{\text{opt}}$ to represent the hypergraph, we treat $\boldsymbol{\rho}_\tau^{[d_{\max}]}$ as data and $\boldsymbol{\rho}_\tau^{[d]}$ as a model of it. The first key aspect of a good model is that it must describe the data as accurately as possible. We can quantify the modeling error, or information loss, with the Kullback–Leibler (KL) entropy divergence between the data and the model, defined as

$$D_{\text{KL}}\left(\boldsymbol{\rho}_\tau^{[d_{\max}]} | \boldsymbol{\rho}_\tau^{[d]}\right) = -S^{[d_{\max}]} + S([d_{\max}]|[d]) \geq 0, \tag{21}$$

where $S^{[d_{\max}]} = -\text{Tr}(\boldsymbol{\rho}_\tau^{[d_{\max}]} \log \boldsymbol{\rho}_\tau^{[d_{\max}]})$ is the Von Neumann entropy of the hypergraph and $S([d_{\max}]|[d]) = -\text{Tr}(\boldsymbol{\rho}_\tau^{[d_{\max}]} \log \boldsymbol{\rho}_\tau^{[d]})$ is the cross-entropy between the hypergraph and its reduced form.

The von Neumann entropy of a hypergraph can also be written as

$$S_\tau^{[d]} = -\text{Tr}\left(\boldsymbol{\rho}_\tau^{[d]} \log \boldsymbol{\rho}_\tau^{[d]}\right) = -\sum_i \lambda_i(\boldsymbol{\rho}_\tau^{[d]}) \log \lambda_i(\boldsymbol{\rho}_\tau^{[d]}), \tag{22}$$

where $\{\lambda_i(\boldsymbol{\rho}_\tau^{[d]})\}$ are the eigenvalues of the density matrix, and $0 \log 0 = 0$ by convention. The von Neumann entropy $S^{[d]}$ is zero if only one eigenvalue is non-zero ("pure state", in the language of quantum mechanics), and maximal equal to $\log N$ if all eigenvalues are equal ("maximally mixed state", in the language of quantum mechanics). This maximal entropy is realized in a trivial hypergraph with $N$ isolated

nodes[42]: all eigenvalues of the Laplacian are zero, so $Z = N$, and $\lambda_i(\boldsymbol{\rho}_\tau^{[d]}) = 1/N \,\forall i$ according to Eq. (18), yielding $S_\tau^{[d]} = \log N$.

Another extreme case is that of the complete hypergraph. In that case, one eigenvalue is $\lambda_1(\mathbf{L}_\tau^{[d]}) = 0$ and the other $N - 1$ are $\lambda_i(\mathbf{L}_\tau^{[d]}) = dN/(N - 1)$ (see Eq. (28) below). The expression for $S_\tau^{[d]}$ can then be derived analytically and, consistent with the case of the complete pairwise network[42], it has minimal entropy $S_\tau^{[d]} = 0$ in the limit $\tau \to +\infty$ and maximal entropy $S_\tau^{[d]} = \log N$ in the limit $\tau \to 0$.

The information loss $D_{\mathrm{KL}}$ takes positive values proportionally to the inaccuracy of the model, and reaches zero at $d = d_{\max}$—as the data is the most accurate model of itself. It is to avoid this overfitting that a regularization term, accounting for model complexity, is needed.

## Model complexity

It is compulsory that high model accuracy must be balanced with low model complexity.

Within the Bayesian framework underlying the minimum description length principle, model complexity arises from the prior term. However, as discussed above, defining a valid prior in terms of the density matrix is nontrivial. Accordingly, we adopt a more heuristic —yet plausible—definition. We measure the complexity of the model in terms of its entropic deviation from the simplest possible model: a network of isolated nodes. We know that the entropy of $N$ isolated nodes is given by $S_{\mathrm{iso}} = \log N$. This gives an upper bound on the von Neumann entropy of a hypergraph, guaranteeing that $0 \leq S_\tau^{[d]} \leq S_{\mathrm{iso}}$. Therefore, we define the model complexity $C$ as:

$$C\left(\boldsymbol{\rho}_\tau^{[d]}\right) = S_{\mathrm{iso}} - S_\tau^{[d]} \geq 0. \tag{23}$$

In fact, by definition we have that the complexity can take values in $C\left(\boldsymbol{\rho}_\tau^{[d]}\right) \in [0, S_{\mathrm{iso}}]$, reaching its minimum for hypergraphs with maximal entropy (e.g., isolated nodes or hypergraphs with high regularity), and its maximum for hypergraphs with minimal entropy (e.g., hypergraphs with more complex structures). From the point of view of the eigenvalues of the density matrix, the model complexity is expected to be lower when the eigenvalues of $\boldsymbol{\rho}_\tau^{[d]}$ are all similar, and larger when they are more diverse.

## Minimizing the cost function

We can now define the cost function $\mathcal{L}$ by combining the information loss in Eq. (21) and the model complexity in Eq. (23):

$$\mathcal{L}\left(\boldsymbol{\rho}_\tau^{[d_{\max}]}|\boldsymbol{\rho}_\tau^{[d]}\right) = D_{\mathrm{KL}}\left(\boldsymbol{\rho}_\tau^{[d_{\max}]}|\boldsymbol{\rho}_\tau^{[d]}\right) + C\left(\boldsymbol{\rho}_\tau^{[d]}\right). \tag{24}$$

By definition, minimizing the cost function corresponds to maximizing the accuracy of the model and, at the same time, minimizing the model complexity (Occam's Razor). To obtain the best compression, we find the smallest order $d$ which gives

$$d_{\mathrm{opt}} = \min_d \mathcal{L}\left(\boldsymbol{\rho}_\tau^{[d_{\max}]}|\boldsymbol{\rho}_\tau^{[d]}\right). \tag{25}$$

It is worth mentioning that the propagation scale $\tau$ works as a resolution parameter. When $\tau$ is very large, the process approaches the steady state, and the network topology becomes irrelevant and, consequently, any model can be a good model. Whereas, at very small $\tau$, the field evolution is linear and through the paths of length $\approx 1$, exhibiting maximum resolution. Although we explore a variety of values of $\tau$, we mainly focus on a characteristic propagation scale $\tau_c$, the largest $\tau$ for which a linearization of the time evolution operator is still valid. Assume the eigenvalues of the Laplacian are given by $\{\lambda_\ell\}$ where $\ell = 1, 2, \ldots N$ and let $\lambda_N$ be the largest of them. Then, the

eigenvalues of the time-evolution operator $e^{-\tau \mathbf{L}}$ are given by $\{e^{-\tau \lambda_\ell}\}$. Here, $\tau_c = 1/\lambda_N$, ensuring that the last eigenvalue of the time evolution operator is reasonably linearizable $e^{-\tau_c \lambda_N} \approx 1 - \tau_c \lambda_N$. This ensures an acceptable linearization for the rest of the eigenvalues since $\lambda_N$ is the largest eigenvalue of the Laplacian.

It is important to remark that in the original message length formulation the hypothesis is that observational data can be described by a generative model characterized by a probability distribution. In turn, such a distribution depends on some unknown parameters that can be fitted by maximizing the Bayesian posterior or, equivalently, minimizing the message length. Consequently, the term accounting for model complexity, $C$, depends on the number of parameters used to describe that probability distribution. Here, we deviate from the standard Bayesian approach, because we work with operators and not with probability distributions, thus a direct generalization of the Bayes theorem in this context is difficult. In fact, our model is a density matrix that characterizes the probability distribution of activating pathways for information flows[60].

Consequently, using our formalism, there is no direct way to reconcile our definition of cost function with the standard Bayesian complexity term. In fact, the latter can be understood as Shannon's surprise about the prior probability, which has no direct counterpart in our formalism. To partially overcome this issue, we use again a Kullback-Leibler divergence to approximate the model complexity and define an effective regularization term. In fact, this reduces to the deviation of Von Neumann entropy of the $d$-th order model from the case of a "gas network" (i.e., $N$ disconnected nodes), which can be understood as the maximally mixed state to define our network of size $N$. In information-theoretical terms, if we define $\boldsymbol{\rho}_{\mathrm{iso}} = \frac{\mathbf{I}}{N}$ as the density matrix of the maximally mixed state, where $\mathbf{I}$ is the identity matrix of order $N$, then $D_{\mathrm{KL}}(\boldsymbol{\rho}|\boldsymbol{\rho}_{\mathrm{iso}}) \equiv C$ encodes the expected "excess surprise" that we have about the state $\boldsymbol{\rho}$ if we use $\boldsymbol{\rho}_{\mathrm{iso}}$ as a prior model for it:

$$
\begin{aligned}
D_{\mathrm{KL}}(\boldsymbol{\rho}|\boldsymbol{\rho}_{\mathrm{iso}}) &= \mathrm{Tr}(\boldsymbol{\rho}(\log \boldsymbol{\rho} - \log \boldsymbol{\rho}_{\mathrm{iso}})) \\
&= \mathrm{Tr}(\boldsymbol{\rho} \log \boldsymbol{\rho}) - \mathrm{Tr}(\boldsymbol{\rho} \log \boldsymbol{\rho}_{\mathrm{iso}}) \\
&= -S - \mathrm{Tr}(\boldsymbol{\rho}(\log \mathbf{I} - \log(N))) \\
&= \log N - S \equiv C.
\end{aligned}
\tag{26}
$$

Remarkably, the term $\log N$ is constant and does not alter the optimization procedure, but it is useful to reconnect—at least in spirit, and not formally—our complexity term $C$ to the surprise about the prior appearing from the classical Bayesian approach. For this reason, throughout this paper, we will use the term cost function instead of description length to acknowledge the existing gap. In this context, alternative definitions can be adopted for both the information loss and the complexity terms. We provide one such example in the Supplementary Information Sec. V and show that, when capturing different features, the resulting outcomes may differ.

## Rescaling $\tau$

**Complete hypergraph.** In some extremely regular structures, such as complete hypergraphs or some simplicial complex lattices, the Laplacians at all orders are proportional. For example, for complete hypergraphs,

$$\mathbf{L}^{(d)} = \frac{K^{(d)}}{N - 1} \mathbf{L}^{(1)}, \tag{27}$$

and consequently

$$\mathbf{L}^{[d]} = \frac{d}{N - 1} \mathbf{L}^{(1)} = d\,\mathbf{L}^{[1]}. \tag{28}$$

Another direct but useful consequence of this is the relationship between the multiorder Laplacians at any two orders

$$\mathbf{L}^{[d]} = \frac{d}{d'}\mathbf{L}^{[d']}. \tag{29}$$

Since by definition Eq. (3), the density matrix $\boldsymbol{\rho}_\tau^{[d]}$ depends only on the product $\tau\mathbf{L}^{[d]}$, we can write

$$\boldsymbol{\rho}_\tau^{[d]} = \frac{e^{-\tau d\mathbf{L}^{[1]}}}{\mathrm{Tr}(e^{-\tau d\mathbf{L}^{[1]}})} = \boldsymbol{\rho}_{\widetilde{\tau}}^{[1]} \tag{30}$$

where $\widetilde{\tau} = d\tau$, or equivalently, between two orders

$$\boldsymbol{\rho}_\tau^{[d]} = \boldsymbol{\rho}_{\frac{d'}{d}\tau}^{[d']}. \tag{31}$$

Hence, instead of selecting a single diffusion time $\tau$ for all orders, we can select a different, more appropriate diffusion time at each order. Specifically, we can select a main $\tau$, and rescale it at each order to ensure that the density matrices are the same. One could choose the rescaling so that the density matrices would be equal to any of them non-rescaled, e.g., to $\boldsymbol{\rho}_\tau^{[1]}$. However, to ensure that the information loss vanishes when considering all possible orders, $D_{\mathrm{KL}}\left(\boldsymbol{\rho}_\tau^{[d_{\max}]}|\boldsymbol{\rho}_\tau^{[d_{\max}]}\right) = 0$, we need to set all of them equal to $\boldsymbol{\rho}_\tau^{[1]}$. This is achieved by rescaling the diffusion time by

$$\tau'(d) = \frac{d_{\max}}{d}\tau \tag{32}$$

so that

$$\boldsymbol{\rho}_{\tau'(d)}^{[d]} = \boldsymbol{\rho}_{\frac{d_{\max}}{d}\tau}^{[d]} = \boldsymbol{\rho}_\tau^{[d_{\max}]}. \tag{33}$$

**General case of proportional Laplacians.** In general, the Laplacian of order $d$ is proportional to that of order 1, $\mathbf{L}^{(d)} \propto \mathbf{L}^{(1)}$, when their respective adjacency matrices are proportional, that is

$$\mathbf{A}^{(d)} = d\,B(d)\mathbf{A}^{(1)}. \tag{34}$$

where $B(d)$ is a coefficient that may depend on the order $d$. Indeed, by definition, the generalized degree and adjacency matrix are related by $K_i^{(d)} = \frac{1}{d}\sum_j A_{ij}^{(d)}$, and thus we also have

$$\boldsymbol{K}^{(d)} = B(d)\boldsymbol{K}^{(1)}, \tag{35}$$

ensuring that the Laplacian matrices are proportional, $\mathbf{L}^{(d)} = B(d)\mathbf{L}^{(1)}$.

Equation (35) implies that

$$B(d) = \langle K^{(d)}\rangle / \langle K^{(1)}\rangle, \tag{36}$$

and hence, the multiorder Laplacian up to order $d$ is given by

$$\mathbf{L}^{[d]} = \frac{d}{\langle K^{(1)}\rangle}\mathbf{L}^{(1)} = d\,\mathbf{L}^{[1]} \tag{37}$$

which is consistent with the complete hypergraph case in Eq. (27), where we have $\langle K^{(1)}\rangle = N - 1$. The rest of the derivation is thus the same as in the complete graph case: rescaling $\tau$ per order as

$$\tau'(d) = \frac{d_{\max}}{d}\tau \tag{38}$$

ensures

$$\boldsymbol{\rho}_{\tau'(d)}^{[d]} = \boldsymbol{\rho}_{\frac{d_{\max}}{d}\tau}^{[d]} = \boldsymbol{\rho}_\tau^{[d_{\max}]}. \tag{39}$$

In turn, this yields no information loss and the flat cost function is given by $\mathcal{L}\left(\boldsymbol{\rho}_\tau^{[d_{\max}]}|\boldsymbol{\rho}_{\tau'}^{[d]}\right) = C\left(\boldsymbol{\rho}_\tau^{[d_{\max}]}\right)\forall d$.

Note again that, in general, a hypergraph does not need to have hyperedges at every order below $D$, unless it is a simplicial complex. If there is no hyperedge at some orders $d$, Eq. (37) must be adjusted, as the factor $d$ comes from the number of orders present. The set of orders present is in general $\mathcal{D} = \{i \leq d : \langle K^{(i)}\rangle > 0\}$, so that Eq. (37) becomes $\mathbf{L}^{[d]} = |\mathcal{D}|\,\mathbf{L}^{[1]}$ and Eq. (38) becomes $\tau'(d) = \frac{d_{\max}}{|\mathcal{D}|}\tau$.

**Higher-order lattices.** For a triangular lattice in which every triangle is promoted to a 2-simplex, each node is part of six 1-simplices and six 2-simplices. Furthermore, each 1-simplex of the lattice is part of two different 2-simplices. This means that each pair of nodes share two 2-simplices if they share one 1-simplex, and zero otherwise. Formally:

$$k_i^{(2)} = 6 = k_i^{(1)} \text{ and } A_{ij}^{(2)} = 2A_{ij}^{(1)}, \tag{40}$$

so that $B(2) = 1$.

### Description of the datasets

We assigned a category to each of the 60 empirical datasets. All datasets are accessible via XGI[26] and stored at https://zenodo.org/communities/xgi/.

*Coauthorship.*—Each of the coauthorship datasets corresponds to papers published in a single year in geology (1980, 1981, 1982, 1983, 1984) and history (2010, 2011, 2012, 2013, 2014). A node represents an author, and a hyperedge represents a publication marked with the "Geology" or "History" tag in the Microsoft Academic Graph[67].

*Contact.*—In the contact datasets, a node represents a person and a hyperedge represents a group or people in close proximity at a given time. Most of the original datasets are from the SocioPatterns collaboration[68,69].

*Drugs.*—The drugs datasets include two constructed in ref. 24 with data from the National Drug Code Directory (NDC). In ndc-classes, a node represents a label (a short text description of a drug's function) and a hyperedge represents a set of those labels applied to a given drug. In ndc-substances, a node represents a substance and a hyperedge is the set of substances in a given drug. The Drug Abuse Warning Network (DAWN) is a national health surveillance system that records drug use that contributes to hospital emergency department visits throughout the United States. A node represents a drug, and a hyperedge is the set of drugs used by a given patient (as reported by the patient) in an emergency department visit. For a period of time, the recording system only recorded the first 16 drugs reported by a patient, so the dataset only uses the first 16 drugs (at most).

*Ecology.*—In the ecology datasets, a node represents a plants species, and a hyperedge represents the set of pollinator species that visit a given plant species. These datasets are from the Web of Life (www.web-of-life.es).

*Email.*—In the email datasets, a node represents an email address and a hyperedge is the set of all recipient addresses included in an email, including the sender's.

*Metabolism.*—Each metabolism dataset is a metabolic reaction networks at the genome scale, from the Biochemical, Genetic and Genomic (BiGG) models database (http://bigg.ucsd.edu/). A node represent a metabolite, and a hyperedge represents a set of

metabolites participating in a metabolic reaction. Different datasets represents different metabolic reaction networks in different organisms. These datasets were analyzed as directed hypergraphs in ref. 70, but we converted them to unweighted undirected hypergraphs.

*Politics.*—In the politics datasets, a node represents a member of the US Congress/House/Senate and a hyperedge is the set of members co-sponsoring a bill between 1973 and 2016. In the House Committee dataset, a node is a member of the US House of Representatives, and a hyperedge is a set of member forming a committee (1993–2017). All datasets were constructed in ref. 24.

*Tags.*—In the tags datasets, a node represents a tag, and a hyperedge is a set of tags associated to a question on online Stack Exchange forums (Mathematics Stack Exchange and Ask Ubuntu). The tag datasets were constructed in ref. 24 with data from the Stack Exchange data dump.

*Theater.*—Each dataset is a Shakespeare theater play, where a node represents a character and a hyperedge represents a set of characters that appear together on stage in a scene. The Hyperbard datasets were introduced in ref. 71.

### Randomizing strategies

We applied three randomization strategies: the configuration model, the node swap, and the shuffling.

The configuration model we used is a generalization of the standard pairwise configuration model to higher-order interactions[25,59]. We applied a configuration model to each order of the hypergraph separately. At each order, the configuration model connects stubs so as to randomize hyperedges while preserving the degree sequences at each order.

The node swap procedure was initially proposed in ref. 17. The idea is to preserve the degree distributions at each order, but to lower the cross-order degree correlations. To do this, at each order $d$, we pair nodes $(i, j)$ and swap their generalized degrees $K_i^{(d)} \leftrightarrow K_j^{(d)}$ by swapping their hyperedge memberships at that order. At each order, each node is paired with a single other nodes so that all nodes are swapped. To more efficiently reduce correlations between orders, high-degree nodes are paired with low-degree degree nodes in priority. Also, to avoid correlation between orders, the node pairing is randomly shuffled at each order.

The shuffling procedure is the most destructive one: it only preserves the number of hyperedges. It works by replacing each $d$-hyperedge by another, randomly selected, non-existing $d$-hyperedge. This procedure destroys intra- and inter-order correlations between hyperedges and tends to produce structures akin to random hypergraphs.

## Data availability

All data is publicly available from XGI-data: https://zenodo.org/communities/xgi/about.

## Code availability

Code for reproducing our results is available online from the repository https://github.com/maximelucas/hypergraph_reducibility and on Zenodo https://doi.org/10.5281/zenodo.17833060[72]. It uses the XGI package[26].

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

## Acknowledgements

M.L. thanks Marco Nurisso for useful feedback on the manuscript and Nicholas Landry for discussions about the configuration model and his publicly available implementations of it and of the simplicial fraction measure. M.L. is a Postdoctoral Researcher of the Fonds de la Recherche Scientifique-FNRS. F.B. and L.G. acknowledge support from the Air Force Office of Scientific Research under award number FA8655-22-1-7025. F.B. acknowledges support from the Austrian Science Fund (FWF) through projects 10.55776/PAT1052824 and 10.55776/PAT1652425. L.G. acknowledges support from the Villum Foundation (project no. 57396) at the University of Copenhagen. M.D.D acknowledge MUR funding within the FIS (DD n. 1219 31-07-2023) Project no. FIS00000158 (CUP C53C23000660001).

## Author contributions

All authors designed the study. M.L., L.G., and A.G. performed the theoretical analysis. M.L. and L.G. performed the numerical simulations. All authors discussed the results. M.L. wrote the original draft and all authors reviewed and edited it. F.B. and M.D.D. supervised the project.

## Competing interests

The authors declare no competing interests.
