## [Transparent Peer Review file · Nature Communications]

Reducibility of higher-order networks from dynamics

Corresponding Author: Dr Maxime Lucas

Version 0:

Reviewer comments:

Reviewer #1

(Remarks to the Author)

I first want to thank and congratulate the authors for addressing my concerns with such care. Since the original version, the authors have made significant improvements to the manuscript and they have resolved my major concerns. In particular, they have :

- compiled the largest dataset of empirical higher-order networks in the field of Network Science and related fields (quality of the data);
- strengthened the narrative by reorganizing the text, adding additional remarks, refining the terminology, and including new references and figures;
- improved the precision of their notation (e.g., in Eqs. (5) and (6));
- added a very insightful analytical result using the hyperring (lines 401 to 425 are particularly illuminating), significantly improving the interpretation of the results in all the paper;

The results clarify how significant are higher-order interactions in complex systems and is definitely moving the field forward (see also my previous comments). After making the minor corrections below, especially meant to improve the flow of the writing, I believe the work is perfectly suited for Nature Communications.

Minor corrections

- 1 - Below Eq. (6), $d_{\text{opt}}(\tau) = d_{\text{max}}$ and $d_{\text{opt}}(\tau) = 1$.
- 2 - Fig 3 caption : it is negative (positive) when $d_{\text{opt}} = 2$ (1) $\rightarrow d_{\text{opt}} = 2$ (= 1) to avoid having (1) like equation (1)
- 3 - Fig 3(b), I did not see a description of the dashed lines and $\tau = 1$ could simply be to the bottom right of the green curve to avoid cluttering
- 4 - 329: \forall \text{all} \rightarrow for all
- 5 - Eq. (11), $\rightarrow \left(\frac{d_{\text{max}}}{d} - 1 \right)$ or $\Big(\Big)$. Suggestion: it would be perhaps more easily readable if, after the equality, you skip the line // so that all terms enter in the second line.
- 6 - Lines 368 is hard to read. In fact, line 363 to 369 should be written more cleanly. It would probably help to write the expectation $E_{\rho} \dots$ and λ_k^{Δ} side by side in one equation $\begin{aligned} \end{aligned}$ after 'where' in line 363, following with their respective description.
- 7 - Lines 391-392 refer to Eqs. (11-12) specifically
- 8 - Line 405: the second expectation should be Δ instead of d
- 9 - There is a contradiction between line 419 and Fig. 3(c) caption. In the former (which I believe is the correct one), it is said that the hyperring is reducible at short time scales while in the latter, it is written that at τ_{short} (where the difference between cost function is negative), $d_{\text{opt}} = 2$ and it is irreducible. Suggestion: 3(c) could be made clearer by increasing its size relative to 3(a,b) (that could be above or on the left side), drawing a grey zone where the difference is negative and writing that $\chi_{\tau}(H) = 0$ in or above this zone.
- 10 - Line 474: in simplicial complex, 'where higher-order hyperedges are strongly nested within lower-order ones' \rightarrow a bit confusing, the sentence is heavy
- 11 - Line 476: add little new spectral ...

(Remarks on code availability)

Reviewer #2

(Remarks to the Author)

Many thanks again for the authors for this substantial round of revisions. I find the new framing of mechanisms vs behaviours much more suitable to the contribution of the paper, and the new analytical hyperring example very illuminating.

I think this manuscript has improved significantly since the original submission, and now I am happy to support its publication in Nature Communications.

However, I must say that two of my concerns still stand (albeit partially addressed):

1) Information flow: I appreciate the authors' efforts to tone down some of their earlier claims around technical terms like "message length" and "information flow". I think this is now more representative of the work itself. Nonetheless, and despite the author's justification around line 155, I think using "information flow" in the title is misleading as it does not involve information flow as used in information theory.

2) Comparisons with Amari's work: I'm afraid I disagree with the somewhat restrictive view of the authors in their reply to my "Comment 6" (as numbered in the response document). Specifically, I disagree with the statement that "these random variables can typically be seen as representing a multivariate time series." As an example of a common application of these measures, one could use a high-order Ising model (with a corresponding network structure), obtain its corresponding $p(x)$, and calculate Amari's measure for each interaction order. This is directly comparable to the authors' work, but for discrete systems. I don't think numerical experiments are necessary, but some discussion of this and other related measures coming from information theory is warranted.

(Remarks on code availability)

Response to Reviewer Reports
NCOMMS-25-87411-T
Reducibility of higher-order networks via information flow

We are grateful to both Reviewers for their positive words and remaining minor comments. We have revised the manuscript accordingly, and provide point-by-point responses below.

Response to Reviewer 1

Comment 1. *I first want to thank and congratulate the authors for addressing my concerns with such care. Since the original version, the authors have made significant improvements to the manuscript and they have resolved my major concerns. In particular, they have :*

- *compiled the largest dataset of empirical higher-order networks in the field of Network Science and related fields (quality of the data);*
- *strengthened the narrative by reorganizing the text, adding additional remarks, refining the terminology, and including new references and figures;*
- *improved the precision of their notation (e.g., in Eqs. (5) and (6));*
- *added a very insightful analytical result using the hyperring (lines 401 to 425 are particularly illuminating), significantly improving the interpretation of the results in all the paper;*

The results clarify how significant are higher-order interactions in complex systems and is definitely moving the field forward (see also my previous comments). After making the minor corrections below, especially meant to improve the flow of the writing, I believe the work is perfectly suited for Nature Communications.

Reply:

We thank the Reviewer for their feedback and positive words. Their feedback since the first round of reviews has pushed us and results in an improved manuscript. We have updated the manuscript to address the minor corrections suggested.

Comment 2.

Minor corrections

- 1 - Below Eq. (6), $d_{\text{opt}}(\tau) = d_{\text{max}}$ and $d_{\text{opt}}(\tau) = 1$.
- 2 - Fig 3 caption : it is negative (positive) when $d_{\text{opt}} = 2$ (1) $\rightarrow d_{\text{opt}} = 2$ (= 1) to avoid having (1) like equation (1)
- 3 - Fig 3(b), I did not see a description of the dashed lines and $\tau = 1$ could simply be to the bottom right of the green curve to avoid cluttering
- 4 - 329: `\forall` \rightarrow for all
- 5 - Eq. (11), $\rightarrow \left(d_{\text{max}}/d - 1\right)$ or $\Big(\Big)$. Suggestion: it would be perhaps more easily readable if, after the equality, you skip the line // so that all terms enter in the second line.
- 6 - Lines 368 is hard to read. In fact, line 363 to 369 should be written more cleanly. It would probably help to write the expectation E_{ρ} ... and λ_k^{δ} side by side in one equation `\begin{align*}` `\end{align*}` after ‘where’ in line 363, following with there respective description.
- 7 - Lines 391-392 refer to Eqs. (11-12) specifically
- 8 - Line 405: the second expectation should be δ instead of d
- 9 - There is a contradiction between line 419 and Fig. 3(c) caption. In the former (which I believe is the correct one), it is said that the hyperring is reducible at short time scales while in the latter, it is written that at τ_{short} (where the difference between cost function is negative), $d_{\text{opt}} = 2$ and it is irreducible. Suggestion: 3(c) could be made clearer by increasing its size relative to 3(a,b) (that could be above or on the left side), drawing a grey zone where the difference is negative and writing that $\chi_{\tau}(H) = 0$ in or above this zone.
- 10 - Line 474: in simplicial complex, ‘where higher-order hyperedges are strongly nested within lower-order ones’ \rightarrow a bit confusing, the sentence is heavy
- 11 - Line 476: add little new spectral ...

Reply:

We have fixed all of these, thank you for the careful reading. For 8., it was actually a typo and should have been d everywhere above—we fixed this. For 9., we fixed the text so that everything is consistent. We also clarified the Figure 3.

Response to Reviewer 2

Comment 1. *Many thanks again for the authors for this substantial round of revisions. I find the new framing of mechanisms vs behaviours much more suitable to the contribution of the paper, and the new analytical hyperring example very illuminating.*

I think this manuscript has improved significantly since the original submission, and now I am happy to support its publication in Nature Communications.

Reply:

We thank the Reviewer for their constructive feedback since the first round of reviews and for their kind words. We have addressed the remaining minor comments.

Comment 2. *However, I must say that two of my concerns still stand (albeit partially addressed):*

1) Information flow: I appreciate the authors' efforts to tone down some of their earlier claims around technical terms like "message length" and "information flow". I think this is now more representative of the work itself. Nonetheless, and despite the author's justification around line 155, I think using "information flow" in the title is misleading as it does not involve information flow as used in information theory.

Reply:

To avoid any confusion, we have changed the title to: "Reducibility of higher-order networks from dynamics".

Comment 3. *2) Comparisons with Amari's work: I'm afraid I disagree with the somewhat restrictive view of the authors in their reply to my "Comment 6" (as numbered in the response document). Specifically, I disagree with the statement that "these random variables can typically be seen as representing a multivariate time series." As an example of a common application of these measures, one could use a high-order Ising model (with a corresponding network structure), obtain its corresponding $p(x)$, and calculate Amari's measure for each interaction order. This is directly comparable to the authors' work, but for discrete systems. I don't think numerical experiments are necessary, but some discussion of this and other related measures coming from information theory is warranted.*

Reply:

We thank the Reviewer for this precision. We have now briefly mentioned this.